# The Effect of Heat Stress on Respiratory Alkalosis and Insulin Sensitivity in Cinnamon Supplemented Pigs

**DOI:** 10.3390/ani10040690

**Published:** 2020-04-15

**Authors:** Jeremy J. Cottrell, John B. Furness, Udani A. Wijesiriwardana, Mitchell Ringuet, Fan Liu, Kristy DiGiacomo, Brian J. Leury, Iain J. Clarke, Frank R. Dunshea

**Affiliations:** 1Faculty of Veterinary and Agricultural Sciences, The University of Melbourne, Parkville VIC 3010, Melbourne, Australia; j.furness@unimelb.edu.au (J.B.F.); udani.wijesiriwardana@unimelb.edu.au (U.A.W.); FLiu@rivalea.com.au (F.L.); kristyd@unimelb.edu.au (K.D.); brianjl@unimelb.edu.au (B.J.L.); fdunshea@unimelb.edu.au (F.R.D.); 2Department of Anatomy and Neurosciences, The University of Melbourne, Parkville VIC 3010, Melbourne, Australia; mringuet@student.unimelb.edu.au; 3Department of Physiology, Monash University, Clayton VIC 3168, Monash, Australia; iain.clarke@unimelb.edu.au

**Keywords:** transepithelial resistance, heat stress, cinnamon, insulin, nutrition, pig, swine, leaky gut

## Abstract

**Simple Summary:**

The response to heat stress (HS) in pigs, like other species includes increasing insulin sensitivity. In this experiment the effects of cinnamon on increasing insulin sensitivity in the HS pig was investigated. The scope of the work was that while cinnamon increased insulin sensitivity overall or in thermoneutral (TN) pigs, no interaction with HS was observed. Consequently, with the exception of intestinal epithelia resistance, no improvement in parameters of HS was observed.

**Abstract:**

With increases in the frequency, intensity and duration of heat waves forecast plus expansion of tropical agriculture, heat stress (HS) is both a current and an emerging problem. As cinnamon has been shown to increase insulin sensitivity, which is part of the adaptive response to HS, the aim of this experiment was to determine if cinnamon could improve insulin sensitivity and ameliorate HS in grower pigs. In a 2 × 2 factorial design, 36 female Large White × Landrace pigs were fed control (0%) vs. cinnamon (1.5%) diets and housed for 7 day under thermoneutral (20 °C, TN) vs. HS conditions (8 h 35 °C/16 h 28 °C, 35% relative humidity). At the completion of the challenge, insulin sensitivity was assessed by an intravenous glucose tolerance test (IVGTT). Heat stress increased parameters such as respiration rate and rectal temperature. Furthermore, biochemical changes in blood and urine indicated the pigs were experiencing respiratory alkalosis. Minimal modelling of parameters of insulin sensitivity showed that HS pigs had a lower insulin response to the IVGTT and improved insulin sensitivity. Cinnamon had additive effects with heat stress, reflected in lowering the insulin area under curve (AUC) and elevated insulin sensitivity compared to TN. However, this apparent improvement in insulin sensitivity did not ameliorate any of the other physiological symptoms of HS.

## 1. Introduction

Higher global temperatures and increased tropical production agriculture mean the challenge of producing livestock under high heat loads is an issue that is increasing in incidence and severity. Pigs are particularly susceptible to heat stress (HS) as they lack eccrine sweat glands [1], compromising evaporative heat loss. Pigs are typically intensively reared in open sided sheds, which provide shade and ventilation but are sensitive to the ambient temperature. While controls such as spray misters and improved shade do provide some benefit, there is an increased focus on nutritional strategies to ameliorate heat stress as an adjunct to existing engineering controls [2].

Pigs reduce feed intake during heat stress, presumably to reduce metabolic heat production. However, counter-intuitively, pigs produced under hot conditions experience increases in adiposity [3,4]. One putative explanation is that this is due to an increase in sensitivity to the anti-lipolytic hormone insulin, which has been observed in pigs [5] and other species [6]. Improved insulin sensitivity may form part of an adaptive response to HS, through improved peripheral blood flow or heat shock protein expression [7]. Reduced insulin sensitivity in diabetic patients and mice has been shown to increase susceptibility to HS [8,9]. A recent meta-analysis has shown that short term cinnamon intake improves glycemic control [10], including in type 2 diabetics [11]. The activity of cinnamon arises from the molecule cinnamonaldehyde, found primarily in *Cinnamomum zeylanicum*, which displays insulinotropic effects via stimulation of the GLUT4 glucose transporter [12] Cinnamon has also been demonstrated to increase insulin sensitivity in grower pigs when used as an in-feed additive [13], although any benefit during HS in pigs has not been investigated. Therefore, the aims of our work were to quantify whether dietary cinnamon increased insulin sensitivity and improved thermal tolerance in HS pigs.

## 2. Materials and Methods

### 2.1. Animals, Diet and Experimental Design

The animal care procedures and the research were approved with protocol 1513462.1 by the University of Melbourne Faculty of Veterinary and Agricultural Sciences Animal Ethics Committee. 

The experiment consisted of a 2 × 2 balanced randomised block design where the factors were experimental diet (control vs. cinnamon) and temperature (TN vs. HS). A total of 36 Large White × Landrace cross-bred female grower pigs (average 41.4 kg) were used in three equal replicates (*n* = 12 pigs/replicate) for the experiment, equating to nine pigs per treatment/group. Female Large white x Landrace were used for the experiment as they are extensively used as finisher pigs in Australia and female pigs are less likely to fight during mixing and transport. Pigs were sourced from a nearby commercial farm and acclimated to experimental diets within the facility for 2 weeks before the experiment. Over this time, pigs were housed in individual pens under TN conditions of a constant 20 °C temperature with 35% humidity. For the climate challenge, pigs were moved to one of two climate control rooms and housed in individual metabolism cages (*n* = 6 room). For the climate challenge one room was maintained at thermoneutrality (TN) while the other room had increased daytime temperatures (09:00 to 17:00) of 35 °C, 35% humidity and overnight temperatures of 28 °C (HS). Pigs were segregated such that the diets in each room were balanced.

The experimental diets were formulated from a wheat-based diets containing either 1.5% cinnamon (derived from *Cinnamomum zeylanicum*) *w/w* vs. 1.5% filler in control diets (Table 1). As HS limits feed intake, the effect of dissimilar feed intake in the TN animals was controlled by limiting feed intake to at 2.5 × maintenance energy, which is the approximate feed intake we have observed during HS in similar experiments [14,15]. This was calculated against body weight and based off National Research Council (NRC) requirements [16]. The daily feed ration was split into two feedings at 09:00 and 15:00.

### 2.2. Indices of Heat Stress

Heat stress was quantified by measuring respiration rate (RR, breaths/min), skin and rectal temperature (ST and RcT) at 09:00, 11:00, 13:00 and 15:00 daily. Skin temperature was recorded on the flank using a hand-held infrared thermometer (Non-contact IR thermometer, Digitech) and rectal temperature quantified using a digital thermometer (Comark PDT 300, Norwich, Norfolk, UK) inserted approximately 2 cm into the anus. Researchers were blinded to the experimental diets and animals were conditioned to handling during the acclimation period.

### 2.3. Intravenous Glucose Tolerance Test

At approximately 08:00 on the last day of the heat period, animals were removed from their rooms and a catheter introduced to the middle or lateral auricular (ear) vein, and advanced so that the catheter tip was situated between the external jugular vein and anterior vena cava [17,18]. To complete this procedure, animals were moved from their metabolism crates into an open space and were restrained in the standing position by using a snout rope and ears were disinfected with 70% ethanol antiseptic solution. An 18 g × 1.25 catheter was introduced to the vein and a guide wire was introduced to a depth of approximately 30 cm. A polyethylene catheter was passed over the wire guide to approximately 35 cm in depth. The wire guide was removed and then the catheter was flushed with heparinised saline (25,000 IU/mL). The catheter was then secured in a pouch and the ear secured by strapping the ear dorsally. The catheter was then used for introducing glucose and sequential sampling of blood during the intravenous glucose tolerance test (IVGTT). At the completion of the IVGTT pigs were euthanised by barbiturate overdose, then dissected to collect ileum samples for immediate analysis with Ussing chambers. Urine samples were collected immediately and directly from the bladder.

For the IVGTT, pigs were first fasted overnight, then, a bolus of 40% dextrose (pre-warmed to 37 °C) was infused over approximately 1 min at a dose of 0.3 g/kg glucose. 5 mL blood samples were withdrawn at −30, −15, −1, 2, 3, 4, 5, 6, 8, 10, 12, 14, 16, 18, 20, 22, 25, 30, 35, 40, 45, 50, 60, 75, 90, 120, 150, 180 and 210 min relative to infusion to quantify glucose, non-esterified fatty acid (NEFA) and insulin kinetics. Once collected, blood samples were immediately placed on ice and promptly centrifuged at 1000× *g* at 4 °C. Plasma was then collected, separated into aliquots and stored at −20 °C until after analysis performed.

### 2.4. Plasma and Urine Analysis

Plasma glucose concentration was quantified following the manufacturer’s instructions using the glucose oxidase method (Infinity^TM^ TR15298, Thermo Scientific) vs. calibrator (Data Cal VA, Thermo Scientific). Plasma NEFA concentrations were quantified using a kit (Cat no. 279-75401, Wako Diagnostics), as previously described [19]. The plasma insulin concentrations were measured in duplicate with an enzyme-linked immunoassay. Briefly, a 96-well plate was coated with anti-insulin antibody raised in guinea pigs (Cat no. gp#603 Antibodies Australia) overnight at 4 °C. The coating solution was de-canted and a blocking solution containing 1.5% bovine serum albumin (BSA) in phosphate buffered saline (PBS) was added at room temperature for 2 h. Once blocked, 10 µL of standard or test samples was added in duplicate with 90 µL of biotinylated anti- body (guinea pig anti-insulin, 1:2500 dilution in 1% BSA/PBS per 5 mM EDTA) and incubated for 2 h at room temperature. Finally, at room temperature streptavidin peroxidase (Cat no. S2438 Sigma-Aldrich, 100 µL of 1: 16000 in 0.1% BSA /PBS per 0.05% Tween 20) was added and incubated for 30 min, followed by 100:l of chromogenic substrate reagent (3, 3, 5, 5 -tetramethyl benzidine, cat # S2438 ThermoFisher Scientific) for 45 min, and the colour reaction was stopped with 100 µL of H_2_SO_4_. The insulin ELISA was calculated to have a sensitivity of 0.2 ng/mL and the following between-assay CV of 16.7% at 2.9 ng/mL, 17.9% at 1.8 ng/mL, 11.7% at 3.3 ng/mL and 13.9% at 1.2 ng/mL [20]. Analysis of plasma electrolyte concentrations and oximetry was performed using a blood gas analyser (Epoc^TM^, Alere, Waltham, MA, USA) with a BGEM test card (10736382, Alere, Waltham, MA, USA) for direct measurement of pH, pCO_2_, pO_2_, Na^+^, K^+^, Ca^++^ and haematocrit (Hct). Urine pH was quantified using digital pH meter (HI991001, HANNA Instruments) and osmolarity quantified (Model 210, Advanced Instruments Inc., Norwood, MA, USA).

### 2.5. Glucose, NEFA and Insulin IVGTT Modelling

Difference in glucose and insulin responses to the IVGTT were quantified using area under the curve analysis (AUC), calculated by the trapezoidal rule. Results of the IVGTT were quantified with minimal modelling using a computer program [21]. Parameters calculated by MINMOD were: basal glucose (G*_b_*); basal insulin (I*_b_*); insulin sensitivity (S*_i_*), which quantifies insulin stimulated disposal by GLUT-4; glucose effectiveness (S*_g_*), which quantifies insulin independent glucose disposal by GLUT-1; acute insulin response to glucose (AIR*_g_*), which reflects the acute insulin secretion and pancreatic responsivity; disposition index (DI = AIR*_g_* × S*_i_*), which combines the parameters of insulin responsiveness and insulin dependent glucose disposal. Paradigm modelling of the homeostatic measures (HOMA) β-cell function and insulin resistance were calculated by MINMOD. A Quantitative Insulin sensitivity Check Index (QUICKI) was calculated from G*_b_*
_and_ I*_b_* (1/log G*_b_* + log I*_b_*), plus a modified QUICKI (mQUICKI) to include basal NEFA (1/log G*_b_* + log I*_b_* + log NEFA_b_) [22]. Paradigm modelling was performed on the average of baseline (−30, −15 and −1 min). Four pigs were excluded from the modelling due to a lack of fit, with the final replication per treatment being nine, six, eight and eight for TN control, TN cinnamon, HS control and HS cinnamon, respectively.

### 2.6. Ussing Chambers

As heat stress disrupts the small intestinal mucosa, sections of jejunum and ileum were collected immediately after euthanasia. Sections were placed in chilled phosphate buffered saline (PBS), then transferred to Krebs solution (11.1 mM glucose, 118 mM NaCl, 4.8 mM KCl, 1.0 mM NaH_2_PO_4_ 1.2 mM MgSO_4_, 25 mM NaHCO_3_, 2.5 mM CaCl_2_, pH 7.4) [23]. The intestine segment was then opened along the mesenteric border and the external muscle was removed. The remaining layers were mounted onto a round slider and placed into a two-part Ussing chamber (EasyMount Diffusion Chambers, Physiologic Instruments) and 5 mL Krebs’ solution added to each side. On the mucosal side, the 11.1 mM glucose was replaced with mannitol, and all solutions were maintained at 37 °C and gassed with 5% CO_2_, 95% O_2_. A multichannel voltage-current clamp (VCC MC6, Physiologic Instruments) was linked to each chamber through a set of four electrodes (two voltage sensing and two current passing electrodes) and agar bridges (3% agarose/3 M KCl in the tip and back filled with 3M KCl) installed on opposite sides of the tissue. Voltage readings were acquired using a PowerLab amplifier and recorded using LabChart^®^5 (Ad Instruments, Sydney, Australia). Tissue was left to equilibrate for 20 min before clamping the voltage to 0 V and epithelial resistance determined by administering five 2 s pulses of 2 mV. The transepithelial resistance (TER) was calculated by Ohm’s law multiplied by the surface area. Afterwards, 200 µL 50 mg/mL FITC-dextran (FD-40) (78331, Sigma, St. Louis, MO, USA) were added into the mucosal side of the chamber, 200 µL solution from both side of a chamber were collected at 1, 30 and 60 min for quantifying mucosa FD-40 apparent permeation coefficient (P_app_) in triplicate by the following equation [24]:P_app_ = dQ/(dt × A × C_0_)
where dQ/dT corresponds to the transport rate in µg/sec, corresponding to the linear slope of the three measures. This was multiplied by the initial concentration in the donor chamber (C_0_, µg/mL) and the area of the slider (0.71 cm^2^).

### 2.7. Intestinal Gas Composition

Intestinal gas was collected from the stomach, jejunum, ileum, and proximal colon using a 1 mL syringe and needle, then injected into an empty 10 mL vacutainer. Blood gas was collected by injecting 1 mL of whole blood into an empty 10 mL vacutainer with 1 mL of 3.5% perchloric acid. After the blood had precipitated, the 1 mL of headspace was removed and injected into a clean 10 mL vacutainer. All vacutainers were then pressurised with helium and transferred to fresh vials for analysis using a 10 mL syringe. The concentrations of CO_2_ and CH_4_ were then quantified relative to standards using an Agilent 7890A Gas Chromatographer with flame ionization detector and electron capture detector. 

### 2.8. Statistical Analysis

Analysis of physiological measures (RR, ST and RcT) was performed by linear mixed model REML for the main effects and interactions of diet (control vs. cinnamon), temperature (TN vs. HS) and time (09:00, 11:00, 13:00, 15:00). The glucose, insulin and NEFA IVGTT time response was analysed by repeated measures REML. For the physiological measures blocking (random factors) was performed on the pig and replicate and for the IVGTT no blocking on experimental replicate was performed and the pig number was used as the subject. The pig and experimental replicates (three in total) were used as a block. The remaining data were analysed with an ANOVA for the main effects and interactions of diet and temperature in Genstat v17 with blocking on the experimental replicate. Normality of results was confirmed by Genstat. The influence of skew and validity of the ANOVA result was confirmed by repeating the analysis against Log_10_ transformed results and validated that non-normality did not contribute to erroneous results. Where main or interactive effects were found to be statistically significant, a post-hoc Tukey’s test was performed and means *p* > 0.05 labelled with different alphabetical characters. For all Figures and Tables, the adjusted means provided by Genstat for the analysis are used and the error presented as the pooled standard error of the difference (SED). 

## 3. Results

Heat stress increased RR, ST and RcT (*p* < 0.001 for all, Figure 1a–c), indicating increased thermoregulatory responses to dissipate excess body heat. Notably the RR and ST plateaued after 4 h of HS (Figure 1a,b). However, RcT, increased by approximately 2 °C in a linear fashion over the 8 h heat period (Figure 1b, Temp × Time *p* < 0.001). No effects of dietary cinnamon were observed on RR, ST or RcT. 

After seven days of TN vs. HS conditions, pigs were fasted, then insulin sensitivity assessed via an IVGTT, presented in Figure 2, with the analysis of the modelling presented in Table 1. Paradigm modelling of fasting (pre-challenge) parameters showed that there was a tendency for basal glucose concentrations to be higher in HS control pigs compared to other groups (*p* = 0.088). Basal insulin (I*_b_***)** concentrations tended to be lower overall in HS pigs (4.98 vs. 3.00 mU/L for TN vs. HS, *p* = 0.052) and fasting NEFA concentrations were lower in HS pigs (369 vs. 216 µmol/L, *p* = 0.006). Additionally, paradigm modelling showed that HS increased QUICKI (0.454 vs. 0.508, *p* = 0.029 for TN vs. HS) and mQUICKI overall (0.496 vs. 0.597, *p* = 0.013). Cinnamon increased QUICKI overall (0.453 vs. 0.508 for control vs. cinnamon, *p* = 0.026), but no interaction was observed. An interactive effect was observed on mQUICKI whereby HS increased insulin sensitivity in the control pigs only (Table 1). There tended to be increased β-cell function in HS pigs (−31.3 vs. −19.2, *p* = 0.058), while no differences in insulin resistance were observed. Glucose storage in the form of liver and muscle glycogen was not influenced by diet or temperature.

The glucose AUC was not influenced by diet or temperature, but there was an additive effect of HS and cinnamon, whereby HS cinnamon pigs had a lower insulin AUC than TN control (Table 2). Overall, cinnamon tended to reduce (17.0 vs. 10.7 mU/L.h, for control vs. cinnamon *p* = 0.062) and HS significantly reduced insulin AUC (17.3 vs. 10.5 mU/L.h, for TN vs. HS, *p* = 0.043). Heat stress reduced the NEFA AUC (1648 vs. 995 nmol/L.h, *p* = 0.004) and no effects or interactions with cinnamon were observed. Glucose effectiveness (S*_g_*) was not influenced by diet or temperature, but insulin sensitivity (S*_i_*) was higher in HS pigs overall (32 vs. 284 mU/L.min for TN vs. HS, *p* = 0.011). The acute insulin response was lower overall in HS pigs than TN (AIR*_g_* 137 vs. 95 mU/L.min, *p* < 0.001). However, in particular the HS control pigs had lower AIR_g_ than their TN counterparts and TN cinnamon pigs. The HS cinnamon pigs were intermediate to the HS controls and TN pigs. The disposition index (DI) was higher in HS than TN pigs (5367 vs. 25737, *p* = 0.043), with no effects or interactions with diet observed.

Pigs exposed to HS tended to have more alkaline blood pH (7.42 vs. 7.43, *p* = 0.087, Table 3), elevated venous pO_2_ (31.9 vs. 38.7 mmHg, *p* = 0.008) and reduced pCO_2_ (53.2 vs. 47.3 mmHg, *p* < 0.001). There was no effect of dietary cinnamon on blood pH and pO_2_, whereas cinnamon reduced pCO_2_ concentrations overall (51.4 vs. 47.1 mmHg, *p* = 0.024), with the effect possibly greater in the HS group (Table 2, *p* = 0.094 for Diet × Temperature). Bicarbonate concentrations were lower in the HS group (34.4 vs. 31.6 mmol/L, *p* = 0.002) and haematocrit reduced by ~17% (31.4 vs. 36.1%, *p* < 0.001), but neither parameter was influenced by diet. A reduction in base excess was observed in HS pigs (8.53 vs. 6.48 mmol/L, *p* = 0.014), indicating alkalosis. Furthermore, there were reductions in blood K^+^ (4.15 vs. 3.96 mmol/L, *p* = 0.036), but not Ca^++^ concentrations. Blood Na^+^ concentrations were not influenced by temperature and electrolytes were not influenced by diet. Urinary pH was more acidic in HS than TN pigs (6.38 vs. 5.61, *p* = 0.001) and the osmolarity was unchanged. Neither urinary pH or osmolarity were affected by diet. 

Ileum barrier function was quantified by the mucosal transepithelial electrical resistance (TER) and apparent permeability (P_app_), respectively (Table 4). Overall, the ileal TER was increased in pigs on the cinnamon diets compared to control (79.6 vs. 90.8 for control vs. cinnamon, *p* = 0.016). Additionally, the TER was higher in HS than TN pigs (80.1 vs. 90.3 for TN vs. HS, *p* = 0.012). There was a significant interaction between diet and temperature, whereby the TER for HS pigs fed cinnamon diets was higher, while all other groups were not significantly different. There were no significant effects of diet or temperature on P_app_, which might be expected to accompany an increased TER.

Gas was collected from the gastrointestinal tract lumen at the stomach, jejunum, ileum, proximal- and distal-colon and analysed for CO_2_ and CH_4_. While no significant effects of cinnamon were observed, there were trends for lower CH_4_ concentrations in the jejunum (*p* = 0.084) and higher CO_2_ in the ileum (*p* = 0.066). By contrast, HS significantly increased ileum CH_4_ concentrations (140 vs. 163, *p* = 0.037 Table 5).

## 4. Discussion

The primary objective of this experiment was to determine whether cinnamon would ameliorate the impacts of HS by increasing insulin sensitivity. The HS protocol resulted in daily increases in evaporative panting and skin temperature that plateaued after 4 h of heat, whereas RcT continued to rise, indicating that the pigs’ thermoregulatory responses were unable to stabilise body temperature. The eight-fold increase in RcT clearly increased respiratory exchange, resulting in CO_2_ to be 12% lower overall. Compensatory reductions in blood bicarbonate of ~8% were observed, while blood pH tended to increase. Overall, these results indicate that the behavioural changes to HS resulted in altered blood biochemistry. “Thermal panting” associated with cooling during HS elevates RR but is coupled with reductions in tidal volume. Despite this adaptation, HS continued to result in hypocapnia, requiring compensatory buffering to prevent respiratory alkalosis. In summary, the pigs in this experiment were clearly in a heat stressed state, as evidenced by an inability to control body temperature and elevated blood biochemical buffering.

We have previously demonstrated that the timeframe and dose of cinnamon supplementation used in this experiment improves insulin sensitivity in thermoneutral pigs, as assessed by an IVGTT [13]. The effects of HS on insulin sensitivity were quantified using paradigm (physiological) and mixed models (IVGTT). These models were in agreement that pigs experiencing HS had increased parameters of insulin sensitivity. This was evidenced by an increased mQUICKI and S*_i_*, respectively, with the latter indicating increased insulin stimulated clearance of glucose by GLUT-4. Non-insulin dependent glucose clearance by GLUT-1 was not influenced by HS (S*_g_*). The acute insulin response (AIR*_g_*) was less in HS pigs than TN, possibly due to increased β-cell function, and HS pigs had lower overall insulin response (AUC) to the glucose challenge. The combination of an increased acute insulin response to glucose and increased insulin sensitivity meant that the disposition index of HS pigs was approximately five-fold higher than TN. Interactions between cinnamon and insulin responsiveness were observed in that there was an improvement in the AIR*_g_* of cinnamon supplemented HS pigs. Furthermore, there was an additive effect between cinnamon and HS on the glucose AUC, whereby HS cinnamon pigs had lower insulin AUCs than TN controls. The mQUICKI result showed that HS increased insulin sensitivity, while the HS cinnamon pigs were intermediate. Collectively these results may indicate that cinnamon does improve insulin sensitivity in HS pigs, but this was not accompanied by improvements in the physiological, metabolic or biochemical responses to HS (Figure 1, Table 3). 

While HS did not influence fasting glucose or insulin concentrations, HS did reduce plasma NEFA concentrations, consistent with previous results from our laboratory [17]. Reduced lipid mobilisation during HS is a characteristic metabolic response, and is thought to contribute to increases in carcass adiposity and potential carcass downgrades in meat species [2,6,25]. As insulin inhibits lipolysis, it has been postulated that HS increases insulin sensitivity. While this has been demonstrated in ruminants [26,27,28], investigations into the effects of HS on pigs has proven to be inconsistent. In a euglycemic clamp model HS increased insulin sensitivity in pigs [5], but in other studies, increased basal insulin [29] and reduced insulin secretion in a stimulated model [30] have been observed, indicating reduced insulin sensitivity. The results of this experiment clearly demonstrate a reduced insulin response following an intravenous glucose challenge, with the insulin response being 40% less than pair fed TN pigs. No differences in the overall glucose response, represented by the AUC, was observed. The apparent improvement in insulin sensitivity was not accompanied by increases in glycogen storage, probably indicating increased glucose utilisation during HS.

Compensatory buffering to prevent alkalosis requires the removal of excess base from the plasma pool. Although excess HCO_3_^−^ can be removed by the kidney and excreted in the urine, this is only likely to occur during acute buffering, as it would equate to a loss of metabolites, and consequently, pigs have a comparatively high threshold for bicarbonate excretion [31]. In the current experiment, HS increased the urinary acidity, consistent with results obtained in cattle following a similar duration of HS [32]. The kidneys are major sites of HCO_3_^−^ metabolism and they exert control by reabsorption of filtered HCO_3_^−^, production of HCO_3_^−^ and excretion of HCO_3_^−^ or acids derived from the diet or metabolism. In terms of acid-base balance in a biological system, excretion of an acid is analogous with production of a base, and therefore, the increasing urinary acidity in this experiment likely reflects efforts to conserve HCO_3_^−^. As the role of renal HCO_3_^−^ production is to neutralise metabolic acid production and maintain pH within normal physiological limits [33], the increasing urinary acidity in this experiment, therefore, likely represents a metabolic acidosis coupled with respiratory alkalosis. This has been characterised to a degree in ruminants, with HS increasing ruminal acidosis [34,35]. The measurement of urinary pH was end-point in this experiment, but dynamic monitoring in HS lactating dairy cows revealed a nyctohemoral pattern where urinary pH increased during hotter daytime temperatures and became more acidic during cooler night time temperatures [36]. This lends support to the approach of supplementing bases in the diet to ameliorate HS. This has been demonstrated in sheep, where sodium hydroxide supplementation ameliorated symptoms of HS [37] and improved layer performance [38]. It is possible that these studies are indirectly benefiting the kidney through a reduced requirement to buffer against metabolic acidosis. 

In the pig, as little as 4–6 h of HS results in desquamation of the ileal epithelial layer and exposure of the lamina propria, which has been shown experimentally to result in reduced TER and increased dextran permeability [39]. In the current experiment, no overall effect of HS on distal ileum FD-40 permeability or TER were observed, indicating that ileum barrier function was not compromised by one week of daily HS. Surprisingly, the cinnamon diet increased ileum TER in the HS pigs. This result requires further validation as it may be due to a variety of factors. The increase in TER may reflect an increased abundance of ion channels, due to an increased mucosal thickness, which runs counter to other publications in the field. There is evidence for cinnamonaldehyde to increase ionic permeability and transport via TRP1A channels [23]. This cannot be excluded, but as the pigs were fasted overnight, it is also unlikely. A subset (12) of the pigs from this experiment were given non-invasive gas sensing capsules that were retained in the stomach and provided real time measurements of lumen CO_2_ and CH_4_ concentrations. In those pigs, we quantified that HS reduced stomach CO_2_ concentrations in the HS pigs [40], which is consistent with reduced rates of metabolic activity. End point analysis of gas headspace in other regions of the GIT in this experiment was highly variable, perhaps as the pigs were fasted before the IVGTT. The result should be viewed with caution but the primary outcome of the analysis was increased CH_4_ concentrations in the terminal ileum, which may reflect an increased microbial population in the terminal ileum as CH_4_ is not normally produced in the small intestine of the pig [41]. 

## 5. Conclusions

Our results show that while the supplementation of cinnamon did not alter physiological responses to HS, cinnamon had additive effects with HS, providing improved insulin sensitivity as evidenced by an increased mQUICKI and S*_i_*. Furthermore, HS pigs supplemented with cinnamon had an improved AIRg and a lower insulin AUC, supporting the findings of an improved insulin sensitivity with cinnamon supplementation. However, dietary cinnamon did not ameliorate the biochemical responses to HS in pigs. These findings support the hypothesis that cinnamon increases insulin sensitivity, but we did not demonstrate any concurrent improvements in thermal tolerance to HS.

## Figures and Tables

**Figure 1 animals-10-00690-f001:**
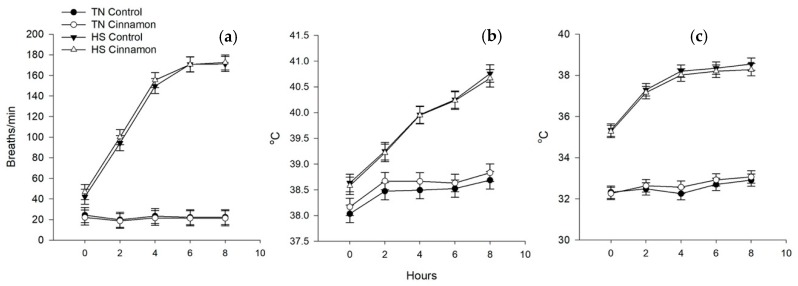
The effects of cinnamon (control vs. 1.5%) supplementation on thermoregulatory responses in pigs housed at 20 °C (thermoneutral, TN) or 35 °C (heat stress, HS) for 7 day. Heat stress increased (**a**) respiration rate, (**b**) rectal and (**c**) skin temperature (*p* < 0.001 for all). No effects of cinnamon on thermoregulatory responses were observed.

**Figure 2 animals-10-00690-f002:**
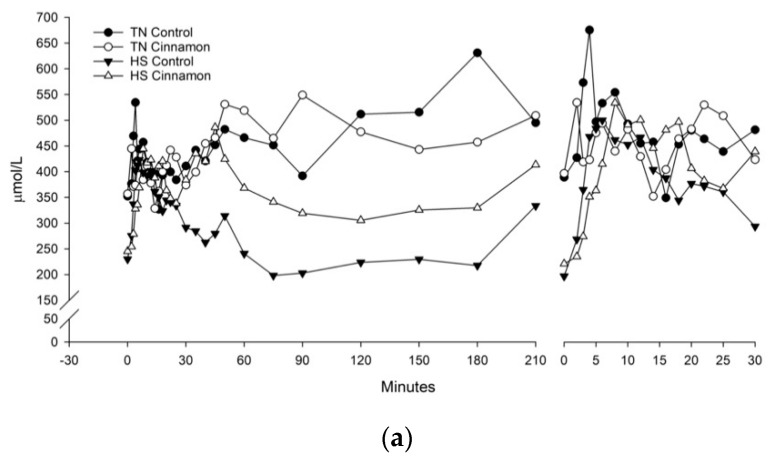
Concentrations of (**a**) glucose, (**b**) NEFA and (**c**) insulin following an intravenous glucose tolerance test (IVGTT) for pigs fed control or 1.5% cinnamon diets and housed under thermoneutral (TN) or heat stress (HS) conditions. On the left-hand side is the full IVGTT result used for MINMOD. On the right-hand side is a truncated view of the first 30 min of the IVGTT. The pooled SEM for glucose, NEFA and insulin were 0.835 mmol/L, 108 µmol/L and 38.1 pmol/L, respectively. The effects of diet, temperature and time were (**a**) *p* = 0.25, 0.59 and <0.001 (**b**) 0.66, 0.104 and 0.063 (**c**) 0.23, 0.082 and <0.001.

**Table 1 animals-10-00690-t001:** Formulation parameters of experimental diets.

Ingredient	%
Wheat	54.0
Lupin kernels 33%	7.25
Mill mix ^1^	13.5
Cinnamon ^2^	1.5
Canola meal 37%	12.0
Meat meal 57%	4.50
Blood meal	1.00
Water	1.00
Phosphorous pre-mix ^3^	0.010
Xylanase ^4^	0.020
Tallow-mixer	3.00
Limestone	1.35
DL-Methionine	0.060
Copper proteinate	0.050
Vitamin premix	0.150
Lysine	0.350
Threonine	0.105
Salt bin micro	0.200
**Calculated values**	
Digestible energy (MJ/kg)	14
Crude Protein (%)	18.3
Lysine (%)	1.40
Leucine (%)	1.33
Calcium (%)	1.07
Total phosphorous (%)	0.642

^1^ Mill mix was reduced to 12% in cinnamon diets; ^2^ for cinnamon diets only; ^3^ NatuPhos 5000, BASF; ^4^ 8000 u/kg xylanase, Porzyme 9310, Dupont Animal Nutrition.

**Table 2 animals-10-00690-t002:** The effects of dietary cinnamon (control vs. cinnamon) and temperature (thermoneutral vs. heat stress) on insulin sensitivity as assessed by intravenous glucose tolerance test (IVGTT) and glycogen storage.

Parameter	Thermoneutral		Heat Stress		SED	*p*-Value		
	Control	Cinnamon	Control	Cinnamon		Diet	Temp.	D × T
Glucose								
AUC (mmol/L h)	19.2	19.2	21.5	18.7	1.62	0.24	0.44	0.23
G*_b_* (mmol/L)	5.65	5.79	6.24	5.65	0.293	0.28	0.28	0.088
S*_g_* (min^−1^)	0.36	0.061	0.040	0.045	0.0153	0.18	0.61	0.39
Insulin								
AUC (mU/L h)	21.6 ^a^	13.1 ^ab^	12.5 ^ab^	8.4 ^b^	4.58	0.062	0.043	0.49
I*_b_* (mU/L)	5.59	4.38	3.15	2.84	1.38	0.44	0.052	0.64
S*_i_* (mU/L min)	53 ^ab^	10 ^a^	165 ^ab^	403 ^b^	128	0.29	0.011	0.13
AIR*_g_* (mU/L min)	142 ^a^	133 ^a^	80 ^b^	109 ^ab^	14.1	0.33	<0.001	0.068
-cell (mU/mM)	−35.1	−27.6	−20.4	−17.9	8.65	0.42	0.058	0.68
Insulin resistance (mM mU L^−2^)	0.079	0.063	0.057	0.040	0.020	0.27	0.14	0.96
DI	5730	5004	12,623	38,851	13,338	0.19	0.043	0.17
NEFA								
Fasting (nmol/L)	382	356	187	245	73.3	0.75	0.006	0.42
AUC (nmol/L h)	1720	1576	763	1227	299	0.46	0.004	0.16
QUICKI	0.439	0.469	0.467	0.548	0.033	0.026	0.029	0.28
mQUICKI	0.445 ^a^	0.547 ^ab^	0.619 ^b^	0.574 ^ab^	0.053	0.22	0.033	0.90
Glycogen (µmol/g)								
Liver	101	118	128	128	16.4	0.63	0.16	0.46
Muscle	152	157	194	171	27.1	0.50	0.14	0.49

NB G*_b_*, S*_g_*, I*_b_, S*_i_*, AIR*_g_*,* β−cell function, insulin resistance and DI calculated by MINMOD. AUC: Area under curve; I*_b_*: basal insulin; S*_i_*: Insulin sensitivity; AIR*_g_*: Acute insulin response to glucose; DI: Disposition index; NEFA: non esterified fatty acids; QUICKI: Quantitative Insulin sensitivity Check Index; mQUICKI: modified Quantitative Insulin sensitivity Check Index.

**Table 3 animals-10-00690-t003:** Effect of dietary cinnamon (control vs. 1.5%) and 7 d thermoneutral vs. heat stress conditions on whole blood and urine biochemistry.

Parameter	Thermoneutral		Heat Stress		SED	*p*-Value		
	Control	Cinnamon	Control	Cinnamon		Diet	Temp.	D × T
Blood								
pH	7.42	7.42	7.43	7.44	0.010	0.51	0.087	0.70
pO_2_ (mmHg)	31.7	32.1	37.0	40.4	3.20	0.42	0.008	0.51
pCO_2_ (mmHg)	53.5	49.3	49.3	45.4	1.28	0.024	<0.001	0.094
HCO_3_ (mmol/L)	34.5	34.3	32.5	30.6	1.09	0.18	0.002	0.27
Haematocrit (%)	31.2	31.5	26.2	26.0	1.38	0.95	<0.001	0.81
Base excess [b] (mmol/L)	8.60	8.45	7.27	5.70	1.07	0.27	0.014	0.36
Na^+^ (mmol/L)	141	138	140	141	1.33	0.34	0.27	0.079
K^+^ (mmol/L)	4.00	3.88	4.30	4.07	0.151	0.12	0.036	0.59
Ca^++^ (mmol/L)	1.28	1.31	1.33	1.31	0.025	0.76	0.22	0.16
Urine								
pH	6.38	6.39	5.39	5.83	0.297	0.29	0.001	0.32
Osmolarity (mOsm)	296	243	308	290	121.8	0.69	0.74	0.84

**Table 4 animals-10-00690-t004:** The effect of dietary cinnamon (control vs. 1.5%) and thermoneutral vs. heat stress conditions on transepithelial resistance (TER) and permeability (FITC) of the small intestine.

Parameter	Thermoneutral		Heat Stress		SED	*p*-Value		
	Control	Cinnamon	Control	Cinnamon		Diet	Temp.	D × T
TER (Ω cm^2^)	80.3 ^a^	79.8 ^a^	78.8 ^a^	101.8 ^b^	6.19	0.016	0.027	0.012
P_app_ (×10^−4^ cm/sec)	174	89	136	124	21.5	0.13	0.96	0.25

**Table 5 animals-10-00690-t005:** Effect of dietary cinnamon (control vs. 1.5%) and thermoneutral vs. heat stress conditions on CO_2_ and CH_4_ concentrations in gas from the gastrointestinal tract.

Parameter		Thermoneutral		Heat Stress		SED	*p*-Value		
		Control	Cinnamon	Control	Cinnamon		Diet	Temp.	D × T
Stomach	CO_2_	5397	3918	4084	4513	908	0.34	0.54	0.15
	CH_4_	156	163	174	140	25.8	0.56	0.94	0.26
Jejunum	CO_2_	9374	4329	3386	5811	3100	0.68	0.23	0.10
	CH_4_	160	133	179	144	24.3	0.084	0.38	0.83
Ileum	CO_2_	3180	8709	2828	4694	2622	0.066	0.26	0.34
	CH_4_	149	130	161	165	14.9	0.57	0.037	0.30
Proximal	CO_2_	8094	8891	7011	6812	1329	0.85	0.12	0.60
Colon	CH_4_	6430	6740	7581	5747	1626	0.47	0.92	0.36
Distal	CO_2_	5578	4844	5839	5992	767	0.59	0.22	0.42
Colon	CH_4_	7510	7233	8420	8920	1370	0.90	0.20	0.69

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
