# Peer review of "The Effect of Heat Stress on Respiratory Alkalosis and Insulin Sensitivity in Cinnamon Supplemented Pigs"

_animals, 2020, doi:10.3390/ani10040690_

Round 1

Reviewer 1 Report

SIMPLE SUMMARY

Line 11: change and other species with linke in other species

Line 13: add the scope of the work

ABSTRACT

Line 20-22: explain more clearly the methods of the work

Line 28: this stands for?

INTRODUCTION

Line 36: delete in Australia and elsewhere

Line 48-51: add some information regarding cinnamon and to which principles contained in cinnamon are due the effects described

Line 52: change this experiment with our work

MATERIALS AND METHODS

Line 57-59: delete the entire phrase and write The animal care procedures and the research were approved with protocol 1513462.1 by the University of Melbourne Faculty of Veterinary and Agricultural Sciences Animal Ethics Committee

Line 61-65: change the order of the sentences. Put first “A total of …for 2 weeks before the experiment” and  then “The experiment …. (TN vs HS)”

Explain   why you use Large whiteX landrance and only female

Line 66: how the pigs were divided in 6 room?

Line 67: add temperature after 20°C

Line 67: add a scheme showing the different group of experimentation

Line 74: add a scheme of the percentage composition of feed

Line 80: correct the hours 0900 with 09:00

Line 108: add after analysis were performed

Line 111: change as per with following

Line 113: add catalogue number of Wako diagnostic

Line 116: add catalogue number of Antibodies Australia

Line 117: this the first time that you write BSA and PBS. Write in full

Line 120: add catalogue information for horseradish peroxidase (industry and catalogue number)

Line 122: add catalogue information for tetramethyl benzidine (industry and catalogue number)

Line 133: delete comma after modeling

Line148: explain the choice of jejum and ileum

Line 151: add the reference for krebs solution

STATISTIC

Did authors check whether there were the conditions to apply linear modeling and ANOVA (i.e. linearity conditions and general conditions for parametric tests)?

RESULTS

Line 258: Fig. 2 (a) the fig. 0-30 min. (at right) has mistakes in the markers.

Lines 271-72: the sentence ”which might be…” could be shifted in the Discussion

DISCUSSION

Line 293: add some information regarding to which principles contained in cinnamon are due the effects described

Line 293: add the scope of the work

Line 293: go ahead with the sentence The HS protocol

Line296: RT?

Line 321: delete in before physiological

Line 322: put in brackets the figure and table indicating the results described

Line 335: TH?

Line 337: change possibly with probably

Line 362: explain why you decide to measure TER

CONCLUSIONS

Line 382: change the results from experiment with our results

Line 386: delete 1 dot before furthermore

Line 386: change furthermore with moreover

Line 387: add after HS pigs the sentence perhaps via either an increase in mucosal thickness or ionic permeability

Line 388-389: delete the sentence This experiment …. Ionic permeability

Author Response

SIMPLE SUMMARY

Line 11: change and other species with linke in other species

Changed to “The response to heat stress (HS) in pigs, like other species includes…

Line 13: add the scope of the work

I’m not sure I understand the comment correctly, but have changed the sentence to:

The scope of the work was that while cinnamon increased insulin…”

ABSTRACT

Line 20-22: explain more clearly the methods of the work

I have moved some of the information regarding the IVGTT from later into the abstract up to the methodology section in lines 22-24.

Line 28: this stands for?

I have amended the final sentence to:

“However, this apparent improvement in insulin sensitivity did not ameliorate any of the other physiological symptoms of HS.”

INTRODUCTION

Line 36: delete in Australia and elsewhere

completed

Line 48-51: add some information regarding cinnamon and to which principles contained in cinnamon are due the effects described

I have inserted the following sentence:

“The activity of cinnamon arises from the molecule cinnamonaldehyde, found primarily in Cinnamonium zeylanicum, which displays insulinotropic effects via stimulation of the GLUT4 glucose transporter [12]”

Line 52: change this experiment with our work

completed

MATERIALS AND METHODS

Line 57-59: delete the entire phrase and write The animal care procedures and the research were approved with protocol 1513462.1 by the University of Melbourne Faculty of Veterinary and Agricultural Sciences Animal Ethics Committee

Completed

Line 61-65: change the order of the sentences. Put first “A total of …for 2 weeks before the experiment” and  then “The experiment …. (TN vs HS)”

In line with comments by reviewer 1 and reviewer 3 I have reordered the sentences in this section. I have started this section with the statement that this experiment was a 2 x 2 factorial design as we believe that this may be causing some confusion.

Explain   why you use Large whiteX landrance and only female

I have added the following sentence:

“Female Large white x Landrace were used for the experiment as they are extensively used as finisher pigs in Australia and female pigs are less likely to fight during mixing and transport.”

Line 66: how the pigs were divided in 6 room?

I have reworded this section to improve the clarity and added the following sentence:

“Pigs were segregated such that the diets in each room were balanced.”

Line 67: add temperature after 20°C

added

Line 67: add a scheme showing the different group of experimentation

Please refer to comment above for lines 61-65.

Line 74: add a scheme of the percentage composition of feed

Added

Line 80: correct the hours 0900 with 09:00

Completed for all

Line 108: add after analysis were performed

completed

Line 111: change as per with following

completed

Line 113: add catalogue number of Wako diagnostic

Added

Line 116: add catalogue number of Antibodies Australia

Added

Line 117: this the first time that you write BSA and PBS. Write in full

Added

Line 120: add catalogue information for horseradish peroxidase (industry and catalogue number)

Added

Line 122: add catalogue information for tetramethyl benzidine (industry and catalogue number)

Added

Line 133: delete comma after modelling

completed

Line148: explain the choice of jejum and ileum

I have added the following to the beginning of the sentence:

“As heat stress disrupts the small intestinal mucosa sections…”

Line 151: add the reference for krebs solution

completed

STATISTIC

Did authors check whether there were the conditions to apply linear modeling and ANOVA (i.e. linearity conditions and general conditions for parametric tests)?

I have added the following sentences:

“Normality of results was confirmed by Genstat. The influence of skew and validity of the ANOVA result was confirmed by repeating the analysis against Log10 transformed results and validated that non-normality did not contribute to erroneous results.”

RESULTS

Line 258: Fig. 2 (a) the fig. 0-30 min. (at right) has mistakes in the markers.

I have updated all three figures as there were inconsistencies with all of them.

Lines 271-72: the sentence ”which might be…” could be shifted in the Discussion

We understand the reviewer but wish to leave in place as these two parameters really need to be considered together. The statement is referring to how the parameters relate to each other, and that is then followed up in the discussion.

DISCUSSION

Line 293: add some information regarding to which principles contained in cinnamon are due the effects described

I have amended this sentence to:

“…..cinnamon would ameliorate the impacts of HS by increasing insulin sensitivity”

I did not change this statement to be more specific such as cinnamonaldehyde or GLUT-4 as these were not specifically measured in this experiment.

Line 293: add the scope of the work

As per earlier the previous comment we are having difficulty understanding if this is an alternative statement to aims or hypotheses.

Line 293: go ahead with the sentence The HS protocol

We’re unclear of what this means. Is it to start the paragraph with this sentence and delete the re-statement of aims?

Line296: RT?

I have changed to RcT

Line 321: delete in before physiological

Deleted

Line 322: put in brackets the figure and table indicating the results described

Figure 1 and Table 2 cited

Line 335: TH?

Changed to “TN”

Line 337: change possibly with probably

Changed

Line 362: explain why you decide to measure TER

We have reworded the sentence as follows:

“In the pig, as little as 4-6 h of HS results in desquamation of the ileal epithelial layer and exposure of the lamina propria, which has been shown experimentally to result in reduced TER and increased dextran permeability”

CONCLUSIONS

Line 382: change the results from experiment with our results

Changed

Line 386: delete 1 dot before furthermore

Deleted

Line 386: change furthermore with moreover

Changed

Line 387: add after HS pigs the sentence perhaps via either an increase in mucosal thickness or ionic permeability

Added

Line 388-389: delete the sentence This experiment …. Ionic permeability

Deleted

Reviewer 2 Report

REPLY FILE: 2020 March 31th: The effect of heat stress on respiratory alkalosis and insulin sensitivity in cinnamon supplemented pigs.

Animals- 764312

--

Dear corresponding Authors,

In general, I found the experiment was properly designed and the data obtained were carefully discussed. The manuscript was logically ordered and was very well written. I value the work that has been done to document the effect of plant phytochemicals on glucose metabolism in response to heat stress in pigs. It is also appreciated that the researchers did the effort to include a thermoneutral control for both dietary treatments and also applied pair-feeding. These results are very valuable to publish.

I only have a few considerations for the authors, mainly related to the measurements on paracellular permeability and TEER. Therefore, I feel that the manuscript can be published after minor revisions. My comments and suggestions will be indicated by line number.

Kind regards and good luck with this manuscript.

MINOR COMMENTS:

--

L 65: Do I understand well that the pigs received the experimental diets (thus either or not including cinnamon) already 2 weeks before HS challenge?

L 73: Would I t be possible to elaborate on the origin of the added cinnamon? Perhaps the authors can include a statement on its cinnamonaldehyde content.

L76: Would it be possible to provide more details on the pair-feeding? Was this done by meal feeding over time, are semi ad libitum?

L 271: To me, it is not that obvious that there would be a link between TEER and FD40 flux. Due to its size, FD40 most likely represents transcellular transport via transcytosis (e.g. 40 kDa HRP). Do the authors speculate on the relation between TEER and tight junction permeability? If so, this would imply that smaller sized molecules, e.g. 4 kDa fluorescein dextrans, need to be used as they allow to reflect on paracellular transport. Furthermore, paracellular passage of a FD4, an uncharged molecule, is controlled by the tight junction proteins occludin and zona occludens-1. This is different from the paracellular passage of small ions, which are rather regulated by claudins.

L278: Would it be possible to confirm the unit of FD40 Papp? 100 à 200 x 10^-4 seems high.

L364:Is it FD4 or FD40. Material and methods indicate 40 kDa FITC.

L 366: Is it correct that an increase in TEER reflex an increase in ionic permeability, or should this be a decrease? Charged solutes can transfer the epithelial mucosa via paracellular transport, and an increase in TEER signifies a lower amount of paracellular transport (in most cases) and thus a decreased ionic permeability. This is different from the effect of phytochemicals on TRP channels, glucose transport across the epithelium and the relation with ion transport as measured by the short circuit current.

L370: Would it be possible to include a hypothesis on how cinnamonaldehyde influences TEER? Receptor-mediated, antioxidant effect, gene expression?

L386: “with cinnamon supplementation. . “ remove the dot.

L389: I would suggest to not include the hypothesis “perhaps via either an increase in mucosal thickness or ionic permeability“ in the conclusion. It might suggest that several different mechanisms were investigated in this study, while in fact this was only studied quite limited (FD40 flux which was not significant).

Author Response

REVIEWER 2

L 65: Do I understand well that the pigs received the experimental diets (thus either or not including cinnamon) already 2 weeks before HS challenge?

This is correct. I have reworded this section, highlighting “experimental diet” rather than just diet and that pigs were acclimated for 2 weeks on experimental diet

L 73: Would I t be possible to elaborate on the origin of the added cinnamon? Perhaps the authors can include a statement on its cinnamonaldehyde content.

We did not assay for cinnamonaldehyde as we did not have the means to do so. I have clarified in the introduction and in the methods that it is the species Cinnamomum zeylanicum that contain cinnamonaldehyde, and that this is the form we used.

L76: Would it be possible to provide more details on the pair-feeding? Was this done by meal feeding over time, are semi ad libitum?

We have reworded this section and I have changed the terminology to restrict fed. As an overview we used NRC values to calculate 2.5 x maintenance energy relative to their body weight.

L 271: To me, it is not that obvious that there would be a link between TEER and FD40 flux. Due to its size, FD40 most likely represents transcellular transport via transcytosis (e.g. 40 kDa HRP). Do the authors speculate on the relation between TEER and tight junction permeability? If so, this would imply that smaller sized molecules, e.g. 4 kDa fluorescein dextrans, need to be used as they allow to reflect on paracellular transport. Furthermore, paracellular passage of a FD4, an uncharged molecule, is controlled by the tight junction proteins occludin and zona occludens-1. This is different from the paracellular passage of small ions, which are rather regulated by claudins.

This is a good comment, they are separate measures based off results from our lab and others (mainly Nicholas Gabler/ Sarah Pearce) there is an increase in FD4 permeability with HS, which has been attributed to impaired tight junction functionality. Our experience is they are not necessarily “coupled” as in this research paper and as you have highlighted.

Regarding the permeability of 4 vs 40 kDa FITC in some upcoming work we have compared 4 kDa FITC and 150 kDa TRITC permeability simultaneously in the neonatal piglet gut in an attempt to quantify gut closure. In that experiment we saw there was less permeability to the larger molecule, but the pattern in the results was exactly the same.

L278: Would it be possible to confirm the unit of FD40 Papp? 100 à 200 x 10^-4 seems high.

There was an error in Table 3 that stated the units at x 104 rather than x 10-4. Due to the larger marker used the apparent permeability is less than other papers on stripped pig mucosa using 4 kDa FITC. For example we have published rates of 20-40 x 10-3 cm/sec (Liu et al., 2016) and Pearce et al (2013) who have units of 2-3 ug/mL/min/cm, which would equate to 300-550 x 10-4cm/sec. We note the reviewers comment, confirm that we have checked the calculations and that we do not have a reason to believe there is an error.

L364:Is it FD4 or FD40. Material and methods indicate 40 kDa FITC.

This has been corrected to FD-40

L 366: Is it correct that an increase in TEER reflex an increase in ionic permeability, or should this be a decrease? Charged solutes can transfer the epithelial mucosa via paracellular transport, and an increase in TEER signifies a lower amount of paracellular transport (in most cases) and thus a decreased ionic permeability. This is different from the effect of phytochemicals on TRP channels, glucose transport across the epithelium and the relation with ion transport as measured by the short circuit current.

This is a very good question, but the way we have looked at this in the paper is that we are studying fasted pigs (they were fasted for the IVGTT). Therefore, while we cannot discount the direct effect of phytochemicals on TEER, the primary driver is most likely to be ion channel abundance, which is driven by mucosal thickness.

I’ve reworded this section to help clarify.

L370: Would it be possible to include a hypothesis on how cinnamonaldehyde influences TEER? Receptor-mediated, antioxidant effect, gene expression?

I have added that cinnamonaldehyde stimulates TRP1A channels

L386: “with cinnamon supplementation. . “ remove the dot.

Removed

L389: I would suggest to not include the hypothesis “perhaps via either an increase in mucosal thickness or ionic permeability“ in the conclusion. It might suggest that several different mechanisms were investigated in this study, while in fact this was only studied quite limited (FD40 flux which was not significant).

This sentence has been deleted

Reviewer 3 Report

I found this paper very interesting and I have no special comments to it.

For me the first part of Material and Methods (2.1) could be more clarify.

The scheme of the study is described well but I lost myself when I was reading it in this part when HS pigs were described. However it is only my feelings but I prefer/recommend presenting the scheme of study in table to show what way authors want to present experimental groups later in a text. Why? - because they are erratic. In figures we can find different presentation than in tables  however I have to underline that it is understandable and the reader is not lost.

My propose is:

The scheme of the study

TN Control

n = 3 (x3)

TN Cinnamon

n = 3 (x3)

HS Control

n = 3 (x3)

HS Cinnamon

n = 3 (x3)

The description in lines 75-76 is not clear for me. I found from cited paper that pigs were fed not totally ad libidum but it was modificated restricted feeding. It means that foe one time pigs had access to part of predicted volume of feed and it was repeated 2.5 times - isn't it? I think it will be better to explain it more clearly in presented paper. in tropic conditions to improve FCR pigs usually had access to feed with higher concentration of energy.

And in the end. In my opinion the first two paragraphs of Discussion are not a discuss it is just summing up and they should be moved to the beginning of the Conclusion.

It is a pity that this experience lasted so short.

Author Response

I found this paper very interesting and I have no special comments to it.

Thank-you!

For me the first part of Material and Methods (2.1) could be more clarify.

The scheme of the study is described well but I lost myself when I was reading it in this part when HS pigs were described. However it is only my feelings but I prefer/recommend presenting the scheme of study in table to show what way authors want to present experimental groups later in a text. Why? - because they are erratic. In figures we can find different presentation than in tables  however I have to underline that it is understandable and the reader is not lost.

My propose is:

The scheme of the study

TN Control

n = 3 (x3)

TN Cinnamon

n = 3 (x3)

HS Control

n = 3 (x3)

HS Cinnamon

n = 3 (x3)

We thank the reviewer for their comments. We believe that the problems may arise from that this experiment is a 2 x 2 factorial design. Please refer to my comments for reviewer 1, we have emphasised this. We hope that this clarification satisfies the reviewers as it is not our preference to present in table form (repetitious and space). We are not sure which part the reviewer found erratic and wonder if it was the presentation of main effects vs interactions?

The description in lines 75-76 is not clear for me. I found from cited paper that pigs were fed not totally ad libidum but it was modificated restricted feeding. It means that foe one time pigs had access to part of predicted volume of feed and it was repeated 2.5 times - isn't it? I think it will be better to explain it more clearly in presented paper. in tropic conditions to improve FCR pigs usually had access to feed with higher concentration of energy.

This has been clarified in line from comments from reviewer 1. The feeding level is calculated to be 2.5 x maintenance energy (ME). A grower pig will consume ~3-4 x ME ad libitum, but the model we and others used will reduce feed intake 30-50%. So to make sure our diet and temperature effects were not skewed by different levels of feed intake the pigs were “pair fed” the same amount of food, whether they were under TN or HS conditions. I have added the following:

“… controlled by “pair feeding”, such that pigs under TN or HS conditions had equivalent feed intake”

And in the end. In my opinion the first two paragraphs of Discussion are not a discuss it is just summing up and they should be moved to the beginning of the Conclusion.

We included those sentences to refresh the reader with the initial objectives of the manuscript.

It is a pity that this experience lasted so short.

We appreciate the sentiment!